# SARS-CoV-2 vaccine uptake in a multi-ethnic UK healthcare workforce: A cross-sectional study

**Christopher A. Martin**[1,2], **Colette Marshall**[3], **Prashanth Patel**[4,5], **Charles Goss**[6], **David R. Jenkins**[7], **Claire Ellwood**[8], **Linda Barton**[9], **Arthur Price**[10], **Nigel J. Brunskill**[5,11], **Kamlesh Khunti**[12,13,14,15]*, **Manish Pareek**[1,2,14,15]*

1 Department of Respiratory Sciences, University of Leicester, Leicester, United Kingdom, 2 Department of Infection and HIV Medicine, University Hospitals of Leicester NHS Trust, Leicester, United Kingdom, 3 University Hospitals of Leicester NHS Trust, Leicester, United Kingdom, 4 Department of Chemical Pathology and Metabolic Diseases, University Hospitals of Leicester NHS Trust, Leicester, United Kingdom, 5 Department of Cardiovascular Sciences, University of Leicester, Leicester, United Kingdom, 6 Department of Occupational Health, University Hospitals of Leicester NHS Trust, Leicester, United Kingdom, 7 Department of Clinical Microbiology, University Hospitals of Leicester NHS Trust, Leicester, United Kingdom, 8 Department of Pharmacy, University Hospitals of Leicester NHS Trust, Leicester, United Kingdom, 9 Department of Haematology, University Hospitals of Leicester NHS Trust, Leicester, United Kingdom, 10 Department of Immunology, University Hospitals of Leicester NHS Trust, Leicester, United Kingdom, 11 Department of Nephrology, Leicester General Hospital, Leicester, United Kingdom, 12 Diabetes Research Centre, Leicester General Hospital, University of Leicester, Leicester, United Kingdom, 13 Leicester Real World Evidence Unit, Diabetes Research Centre, Leicester General Hospital, University of Leicester, Leicester, United Kingdom, 14 NIHR Leicester Biomedical Research Centre, Leicester, United Kingdom, 15 NIHR Applied Research Collaboration East Midlands, Leicester, United Kingdom

* kk22@leicester.ac.uk (KK); mp426@leicester.ac.uk (MP)

**Data Availability Statement:** The cohort was extracted under Caldicott Guardian approval for a specific purpose and as part of our undertaking

## Abstract

### Background

Healthcare workers (HCWs) and ethnic minority groups are at increased risk of COVID-19 infection and adverse outcomes. Severe acute respiratory syndrome coronavirus 2 (SARS-CoV-2) vaccination is now available for frontline UK HCWs; however, demographic/occupational associations with vaccine uptake in this cohort are unknown. We sought to establish these associations in a large UK hospital workforce.

### Methods and findings

We conducted cross-sectional surveillance examining vaccine uptake amongst all staff at University Hospitals of Leicester NHS Trust. We examined proportions of vaccinated staff stratified by demographic factors, occupation, and previous COVID-19 test results (serology/PCR) and used logistic regression to identify predictors of vaccination status after adjustment for confounders. We included 19,044 HCWs; 12,278 (64.5%) had received SARS-CoV-2 vaccination. Compared to White HCWs (70.9% vaccinated), a significantly smaller proportion of ethnic minority HCWs were vaccinated (South Asian, 58.5%; Black, 36.8%; $p < 0.001$ for both). After adjustment for age, sex, ethnicity, deprivation, occupation, SARS-CoV-2 serology/PCR results, and COVID-19-related work absences, factors found to

with them we are not to further routinely share this data. The data are held in an institutional repository. Interested parties, with appropriate approvals, can apply for data access by contacting the R&I Chief Operating Officer at UHL (R&IAdmin@uhl-tr.nhs.uk). Reasonable requests will be assessed on a case-by-case basis in discussion with the Caldicott Guardian.

**Funding:** CAM is an NIHR academic clinical fellow (ACF-2018-11-004). KK is supported by NIHR Applied Research Collaboration East Midlands (ARC EM). KK and MP are supported by the NIHR Leicester Biomedical Research Centre (BRC). MP is supported by a NIHR Development and Skills Enhancement Award (NIHR301192) and funding from UKRI/MRC (MR/V027549/1). The views expressed are those of the author(s) and not necessarily those of the NIHR, NHS or the Department of Health and Social Care. The funders had no role in design, data collection and analysis, decision to publish, or preparation of the manuscript.

**Competing interests:** I have read the journal's policy and the authors of this manuscript have the following competing interests: MP reports grants and personal fees from Gilead Sciences and personal fees from QIAGEN, outside the submitted work. KK is a member of Independent SAGE and the ethnicity subgroup of SAGE and national lead for ethnicity and diversity for National Institute for Health Applied Research Collaborations and Director for University of Leicester Centre for Black Minority Ethnic Health.

**Abbreviations:** aOR, adjusted odds ratio; ESR, Electronic Staff Record; HCW, healthcare worker; IMD, Index of Multiple Deprivation; OR, odds ratio; SARS-CoV-2, severe acute respiratory syndrome coronavirus 2; UHL, University Hospitals of Leicester NHS Trust.

be negatively associated with vaccine uptake were younger age, female sex, increased deprivation, pregnancy, and belonging to any non-White ethnic group (Black: adjusted odds ratio [aOR] 0.30, 95% CI 0.26–0.34, $p < 0.001$; South Asian: aOR 0.67, 95% CI 0.62–0.72, $p < 0.001$). Those who had previously had confirmed COVID-19 (by PCR) were less likely to be vaccinated than those who had tested negative. Limitations include data being from a single centre, lack of data on staff vaccinated outside the hospital system, and that staff may have taken up vaccination following data extraction.

## Conclusions

Ethnic minority HCWs and those from more deprived areas as well as younger staff and female staff are less likely to take up SARS-CoV-2 vaccination. These findings have major implications for the delivery of SARS-CoV-2 vaccination programmes, in HCWs and the wider population, and should inform the national vaccination programme to prevent the disparities of the pandemic from widening.

## Author summary

### Why was this study done?

- Healthcare workers, particularly those from ethnic minority groups, are at high risk of COVID-19.

- There are concerns that uptake of vaccination against COVID-19 in healthcare workers may vary by ethnicity as well as other demographic, occupational, and health factors, but there is limited real-world evidence on this topic.

- Determining factors that are associated with a lack of vaccine uptake in healthcare workers is important as it allows for targeted interventions to improve vaccine uptake, which will protect healthcare workers and the patients under their care.

### What did the researchers do and find?

- We used routinely collected data from a hospital vaccination programme to establish which staff at a large, ethnically diverse hospital trust in the UK had accepted the offer of vaccination against COVID-19.

- We combined these data with data on the demographic and occupational characteristics of staff members and also with data on previous test results and work absences for COVID-19.

- Using this dataset, we were able to determine that 65% of staff had accepted vaccination and that vaccine uptake was significantly lower in ethnic minority groups, younger age groups, females, pregnant healthcare workers, those living in more deprived areas, and those with a history of COVID-19.

**What do these findings mean?**

- Our findings indicate that there are many healthcare workers who have not accepted a vaccine against COVID-19, which has important implications for the risk of infection for the individual healthcare workers and for patients under their care.

- We have identified particular demographic and occupational groups that should be targeted for interventions aimed at improving vaccine uptake.

- To make these interventions effective, more research should be undertaken to understand what the barriers are to COVID-19 vaccination in these groups and to evaluate methods of overcoming these barriers.

## Introduction

COVID-19, the disease caused by infection with severe acute respiratory syndrome coronavirus 2 (SARS-CoV-2), has spread to become a global pandemic causing significant morbidity and mortality in many countries. As of February 2021, total worldwide COVID-19 cases are estimated to be over 100 million, and deaths related to COVID-19 number over 2.1 million [1]. In recent months, thanks to an unprecedented global research effort, a number of vaccines against SARS-CoV-2 have been developed and approved [2,3], and it is hoped that mass vaccination programmes will aid in slowing transmission of the virus as well as reducing hospitalisation and death from COVID-19.

As the pandemic has progressed, it has become clearer that certain factors may increase the risk of acquiring SARS-CoV-2 infection, including age, obesity and the presence of particular comorbidities (e.g., diabetes and cardiovascular disease), occupation, and household size [4–6]. Amongst these 'high-risk' groups are healthcare workers (HCWs) [6,7], in whom an increased risk of hospitalisation with COVID-19 has also been demonstrated [8]. Within a HCW population, it has been shown that the risk of SARS-CoV-2 infection differs by occupational role and is highest in 'front-door' and patient-facing specialities [8,9], implying that at least some of the increased risk faced by HCWs is mediated through occupational exposure to SARS-CoV-2. In recognition of this risk, the UK Joint Committee on Vaccination and Immunisation (JCVI) listed frontline HCWs as a priority group for receiving vaccination against SARS-CoV-2 [10].

The COVID-19 pandemic has also disproportionately affected those from ethnic minority groups, with previous work demonstrating an increased risk of SARS-CoV-2 infection and adverse outcomes relative to White individuals [5,11,12]. Furthermore, HCWs of minority ethnicity have been shown to be at higher risk of infection than their White colleagues [9,13].

In light of the increased risk of COVID-19 infection and adverse outcomes faced by ethnic minority HCWs, concerns have been raised regarding uptake of the SARS-CoV-2 vaccine in this group, both in the UK and in the US [14,15]. These concerns are founded upon previous work conducted in the general population, which has demonstrated reduced vaccine uptake by ethnic minority individuals [14], as well as recent survey studies investigating intentions to receive vaccination against COVID-19, which have demonstrated an increased likelihood of SARS-CoV-2 vaccine hesitancy in ethnic minority groups, including amongst HCWs [15–17].

SARS-CoV-2 vaccination in a HCW cohort is important not only for protection of the individual but, given that a significant proportion of COVID-19 inpatients acquire their infection

in hospital and that this has been attributed to HCW-to-patient transmission, may also prove to be important for reducing nosocomial transmission of COVID-19 [18,19]. Despite HCWs being important targets for vaccination, there are few studies examining SARS-CoV-2 vaccine uptake (as opposed to vaccine intention) amongst HCWs. A small study of staff working in a specialist orthopaedic hospital in London, UK, determined vaccine uptake to be 62% [20]. A larger nationwide cohort study in UK HCWs determined overall vaccine uptake to be 89% as of 5 February 2021, although, as the authors acknowledge, this estimate may not be generalisable due to the potential for self-selection bias in a consented cohort study [21]. Both studies found lower vaccine uptake amongst those from minority ethnic groups (compared to those from White ethnic groups), females, and those in nursing and portering/estates roles. Additionally, it has been suggested that UK HCWs with a history of COVID-19 may be less likely to take up vaccination than those without evidence of previous SARS-CoV-2 infection [21]. Data from outside the UK are limited; an Israeli study reported SARS-CoV-2 vaccine uptake in HCWs to be 90% but did not attempt to determine factors associated with uptake [22].

To add to the limited evidence base on this important public health issue, particularly in light of the potential requirement for a SARS-CoV-2 booster vaccination for UK HCWs, we sought to determine the effects of demographic factors (including ethnicity), occupational factors, and previous COVID-19 on SARS-CoV-2 vaccine uptake in a large multi-ethnic UK healthcare workforce.

## Methods

This study is reported as per the Strengthening the Reporting of Observational Studies in Epidemiology (STROBE) guideline (S1 Checklist).

### Study design and study centre

This cross-sectional surveillance was conducted at University Hospitals of Leicester NHS Trust (UHL), one of the largest acute hospital trusts in the UK, where 36% of staff are from minority ethnic backgrounds [23]. UHL is the only acute hospital trust serving the population of Leicester, Leicestershire, and Rutland (approximately 1 million residents) and cares for the vast majority of hospital attenders with COVID-19 from these areas. Leicester has seen comparatively high rates of SARS-CoV-2 transmission across the course of the pandemic compared to other areas of the UK and was subject to extended 'lockdown' measures in June and July 2020 [24,25].

### Staff vaccination programme

UHL began vaccinating staff against SARS-CoV-2 on 12 December 2020, initially using the BNT162b2 mRNA (Pfizer–BioNTech) COVID-19 vaccine [2] and subsequently, from 8 January 2021, also the ChAdOx1 nCoV-19 (Oxford–AstraZeneca) COVID-19 vaccine [3]. Staff were not given a choice of which vaccine they were offered. Three vaccination 'hubs' (1 at each of the 3 main hospital sites that make up UHL) were established on 12 December 2020, 8 January 2021, and 15 January 2021.

At the launch of the vaccine programme, priority was given to those who were 'clinically extremely vulnerable' (i.e., those at highest risk of severe COVID-19), working in high-exposure areas, or working with the most vulnerable patient groups. Over the following weeks, priority was extended to all staff over 50 years of age or with 2 or more vulnerability factors [26]. Between 7 and 20 January 2021, all patient-facing staff were invited to attend. By the end of this period, there was frequently surplus capacity, and invitations were extended to any registered health or social care worker in the region (i.e., capacity was such that vaccinations were

offered even to those outside the acute hospital trust). All staff at UHL received an email inviting them to attend for vaccination and also received regular reminders to book vaccination appointments via trust-wide electronic and verbal cascaded communications. Line managers were instructed to publicise vaccination, particularly in areas where there is a known low rate of internet or smart phone usage by staff. Vaccine hubs at UHL were well resourced, and, to our knowledge, there were no instances during the study period of vaccination appointments being postponed due to a lack of capacity. Furthermore, as vaccination capacity increased with the addition of vaccine hubs, drop-in sessions were made available to staff.

### Study population

We included all staff identified in the Electronic Staff Record (ESR)—which encompasses all permanent, part-time, locum, and bank workers employed by UHL—on 3 February 2021.

### Data collection

**Outcome variable.** The outcome was uptake of at least 1 dose of SARS-CoV-2 vaccine. This was established by extracting data from the National Immunisation and Vaccination System (NIVS).

**Covariates.** We extracted information concerning age (categorised into groups of ≤30, 31–40, 41–50, 51–60, and ≥61 years old), sex, occupational role (categorised into 7 groups—S1 Table), and residential postcode from the ESR. We used residential postcode to obtain the Index of Multiple Deprivation (IMD) quintile using an online tool provided by the UK government. IMD is the official measure of relative deprivation for small areas of the UK [27].

We also collected data on self-reported ethnicity (this was categorised into White, South Asian, Black, and Other for the main analysis, and into a larger number of ethnicity categories based on the 18 categories used by the UK's Office for National Statistics [28] for a more granular subanalysis—S2 Table). Ethnicity has previously been defined as 'the social group a person belongs to, and either identifies with or is identified with by others, as a result of a mix of cultural and other factors including language, diet, religion, ancestry, and physical features traditionally associated with race' [29].

We used data from occupational health records to determine the number, date, and result of any SARS-CoV-2 polymerase chain reaction (PCR) or anti-SARS-CoV-2 serology tests as well as the reason given for any recorded COVID-19-related absences from work since the start of the pandemic.

### Data analysis

An analysis plan was generated prospectively through discussion with all co-authors prior to accessing the data. These discussions informed the original draft of the methods section of the paper, which was prepared prior to data extraction. Dependent variables that may be associated with the outcome (and are routinely collected through the ESR) were selected a priori with reference to existing literature. Beyond the collapsing of categories of certain demographic variables (granular ethnicity categories and occupational roles) due to low numbers of HCWs in some categories, there were no data-driven changes to the analyses conducted.

All variables were categorical and were summarised as count and percentage. We tested differences between vaccinated and unvaccinated cohorts using chi-squared tests. The number and percentage of staff vaccinated in each week from the start of the vaccination programme to the date of data extraction were plotted.

We used logistic regression to evaluate the univariable association of age, sex, ethnicity, deprivation, occupation, SARS-CoV-2 serology and PCR results, and the reason given for any

COVID-19-related work absences (including symptomatic infection, household infection, or pregnancy) with SARS-CoV-2 vaccine uptake and present the results as odds ratios (ORs) with 95% confidence intervals (95% CIs). We also used multivariable logistic regression to determine adjusted ORs (aORs) and 95% CIs after adjustment for age, sex, ethnicity, deprivation, occupation, SARS-CoV-2 serology and PCR results, and the reason given for any COVID-19-related work absences.

Multiple imputation was used to replace missing data in all logistic regression models; the multiple imputation model included all variables bar those being imputed. Rubin's rules were used to combine the parameter estimates and standard errors from 10 imputations into a single set of results [30].

All analyses were conducted using Stata Statistical Software (release 16.1; StataCorp, College Station, TX). $p$-Value $< 0.05$ was considered statistically significant. Figures were prepared in Excel (Microsoft).

## Ethics

We consulted the NHS Health Research Authority decision aid to ascertain whether ethical approval was required. It was deemed that, as this work represents a service evaluation/surveillance that utilises data collected as part of the routine delivery of a clinical service, approval was not required. In addition, we confirmed approval from our Caldicott Guardian to undertake this work as an audit (UHL11113).

## Results

### Demographic and occupational characteristics of the cohort

In total, 19,044 HCWs were included in the final analysis (see Table 1). In total, 47.7% were under 40 years of age and 75.6% were female; 60.3% were White, 25.5% were South Asian, and 7.1% were Black. Data were missing for 110 values for IMD and 1 value for date of vaccination. The number of vaccinations per week peaked in the week 11–17 January 2021 and has been in decline since (Fig 1).

### Impact of demographic and occupational factors on SARS-CoV-2 vaccination uptake

In total, 64.5% of HCWs received SARS-CoV-2 vaccination during the study period. Unvaccinated HCWs were younger than vaccinated HCWs (31.7% of unvaccinated HCWs were ≤30 years old compared to 18.7% of the vaccinated cohort [$p < 0.001$]).

Compared to White HCWs (70.9% vaccinated), a significantly lower proportion of ethnic minority HCWs were vaccinated (South Asian, 58.5%; Black, 36.8%; $p < 0.001$ for both; see Table 1 and Fig 1). Within the South Asian cohort, a significantly smaller proportion of Pakistani and Bangladeshi HCWs were vaccinated compared to the Indian cohort (43.2% and 36.8% versus 60.3%, respectively; $p < 0.001$ for both comparisons). The proportions of vaccinated Black Caribbean and Black African HCWs were similar (39.7% versus 36.2%; $p = 0.32$; S3 Table).

The unvaccinated cohort had a greater proportion than the vaccinated cohort of HCWs living in areas corresponding to the lower 3 IMD quintiles (61.4% versus 50.8%; $p < 0.001$; Table 1).

The occupational groups with the lowest proportions of vaccinated HCWs were doctors (57.4%), estates and facilities staff (60.7%), and nurses, midwives and HCAs (62.5%). The occupational group with the highest proportion of vaccinated staff comprised those in

**Table 1. Description of the cohort by vaccination status.**

| Variable | Total, n = 19,044 | Unvaccinated, n = 6,766 (35.5%) | Vaccinated, n = 12,278 (64.5%) |
|---|---|---|---|
| **Age (years)** | | | |
| ≤30 | 4,432 (23.3) | 2,142 (31.7) | 2,290 (18.7) |
| 31–40 | 4,656 (24.5) | 1,975 (29.2) | 2,681 (21.8) |
| 41–50 | 4,312 (22.6) | 1,275 (18.8) | 3,037 (24.7) |
| 51–60 | 4,101 (21.5) | 975 (14.4) | 3,126 (25.5) |
| ≥61 | 1,543 (8.1) | 399 (5.9) | 1,144 (9.3) |
| **Sex** | | | |
| Female | 14,395 (75.6) | 5,099 (75.4) | 9,296 (75.7) |
| Male | 4,649 (24.4) | 1,667 (24.6) | 2,982 (24.3) |
| **Ethnicity** | | | |
| White | 11,485 (60.3) | 3,338 (49.3) | 8,147 (66.4) |
| South Asian | 4,863 (25.5) | 2,020 (29.9) | 2,843 (23.2) |
| Black | 1,357 (7.1) | 858 (12.7) | 499 (4.1) |
| Other | 1,038 (5.4) | 429 (6.3) | 609 (5.0) |
| Not stated | 301 (1.6) | 121 (1.8) | 180 (1.5) |
| **IMD quintile** | | | |
| 5 (least deprived) | 4,597 (24.3) | 1,323 (19.6) | 3,274 (26.7) |
| 4 | 4,010 (21.2) | 1,265 (18.7) | 2,745 (22.4) |
| 3 | 3,302 (17.4) | 1,175 (17.4) | 2,127 (17.3) |
| 2 | 4,085 (21.6) | 1,682 (24.8) | 2,403 (19.6) |
| 1 (most deprived) | 2,940 (15.5) | 1,252 (18.5) | 1,688 (13.8) |
| Missing | 110 (0.6) | 69 (1.0) | 41 (0.3) |
| **Occupation** | | | |
| Doctor | 3,001 (15.8) | 1,280 (18.9) | 1,721 (14.0) |
| Nurse/midwife/HCA | 7,815 (41.0) | 2,929 (43.3) | 4,886 (39.8) |
| Allied health professional | 1,380 (7.3) | 427 (6.3) | 953 (7.8) |
| Administrative/executive | 3,465 (18.2) | 928 (13.7) | 2,537 (20.7) |
| Healthcare scientist | 871 (4.6) | 237 (3.5) | 634 (5.2) |
| Estates/facilities | 2,306 (12.1) | 907 (13.4) | 1,399 (11.4) |
| Other | 206 (1.1) | 58 (0.9) | 148 (1.2) |
| **Previous SARS-CoV-2 serology** | | | |
| Never tested | 7,456 (39.2) | 3,656 (54.0) | 3,800 (31.0) |
| Negative | 10,314 (54.2) | 2,732 (40.4) | 7,582 (61.8) |
| Positive | 1,274 (6.7) | 378 (5.6) | 896 (7.3) |
| **Previous SARS-CoV-2 PCR** | | | |
| Never tested | 15,136 (79.5) | 5,710 (84.4) | 9,426 (76.8) |
| Negative | 3,072 (16.1) | 761 (11.3) | 2,311 (18.8) |
| Positive | 836 (4.4) | 295 (4.4) | 541 (4.4) |
| **Previous COVID-19 work absence** | | | |
| No absence | 12,619 (66.3) | 4,749 (70.2) | 7,870 (64.1) |
| Symptomatic | 3,698 (19.4) | 1,221 (18.1) | 2,477 (20.2) |
| Household or test and trace contact | 2,727 (14.3) | 796 (11.8) | 1,931 (15.7) |
| Pregnant | 130 (0.7) | 106 (1.6) | 24 (0.2) |

All data presented as n (%).

COVID-19, coronavirus disease 2019; HCA, healthcare assistant; IMD, Index of Multiple Deprivation; PCR, polymerase chain reaction; SARS-CoV-2, severe acute respiratory syndrome coronavirus 2.

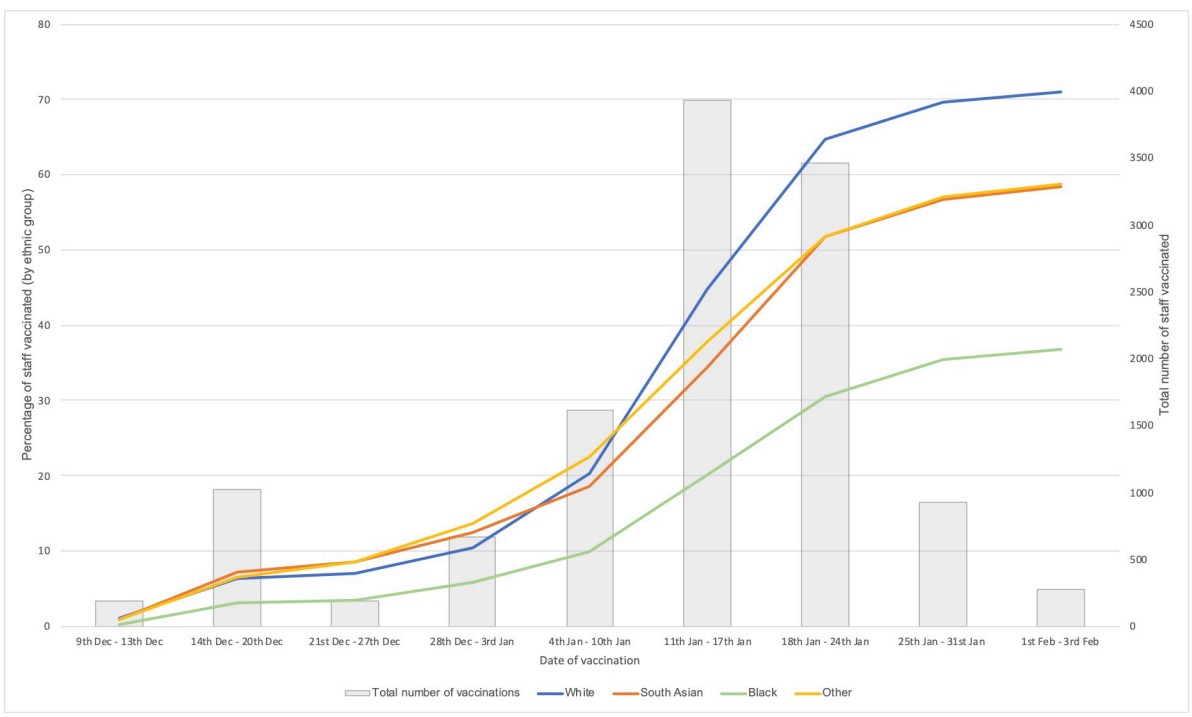

**Fig 1. Number and percentage of staff vaccinated over time by ethnic group.** The figure shows the number of staff vaccinated (grey bars) and the cumulative percentage of the total number of staff of each ethnic group vaccinated (coloured lines) each week since the start of the University Hospitals of Leicester NHS Trust vaccination programme. It should be noted that the first and last time points do not represent complete weeks. There was 1 missing value for date of vaccination, and this observation is excluded from the figure.

administrative and executive roles (73.2%; see Table 1). In a sensitivity analysis excluding those with locum or bank contracts, vaccination uptake amongst doctors was higher (69.4%; S4 Table). A more granular analysis of vaccination uptake in medical staff is shown in S5 Table. Within medical staff, vaccination uptake was highest amongst consultants (81.5% vaccinated) and foundation year 1 doctors (69.6% vaccinated) and lowest amongst senior house officers/speciality registrars (42.5% vaccinated), medical support staff (31.9% vaccinated), and the small number of general practitioners (9.5% vaccinated).

## Association of prior SARS-CoV-2 infection with vaccination uptake

In total, 11,588 (60.8%) staff had previously been tested for anti-SARS-CoV-2 IgG (11.0% of these tests were positive), and 3,908 (20.5%) had previously undergone nasopharyngeal PCR testing for SARS-CoV-2 (21.4% of these tests were positive).

When only staff who had undergone serological testing were analysed, staff with detectable antibody formed a greater proportion of the unvaccinated cohort compared to the vaccinated cohort (12.2% versus 10.6%; $p$ = 0.02). Proportions of those with a previous positive SARS-CoV-2 PCR test in the vaccinated and unvaccinated cohorts were the same (4.4% for both).

## Factors associated with SARS-CoV-2 vaccine uptake

Table 2 shows univariable and multivariable (adjusted for age, sex, ethnicity, IMD, occupation, previous SARS-CoV-2 testing, and COVID-19-related work absences) logistic regression models for factors associated with vaccination against SARS-CoV-2.

**Table 2. Univariable and multivariable analysis of factors associated with SARS-CoV-2 vaccine uptake.**

| Variable | n vaccinated/N total (%) | OR (95% CI) | p-Value | aOR (95% CI) | p-Value |
|---|---|---|---|---|---|
| **Age (years)** | | | | | |
| ≤30 | 2,290/4,432 (51.7) | 0.45 (0.41–0.49) | <0.001 | 0.48 (0.44–0.53) | <0.001 |
| 31–40 | 2,681/4,656 (57.6) | 0.57 (0.52–0.62) | <0.001 | 0.64 (0.58–0.70) | <0.001 |
| 41–50 | 3,037/4,312 (70.4) | Reference | <0.001 | Reference | — |
| 51–60 | 3,126/4,101 (76.2) | 1.35 (1.22–1.48) | <0.001 | 1.19 (1.07–1.31) | 0.001 |
| ≥61 | 1,144/1,543 (74.1) | 1.20 (1.06–1.37) | <0.001 | 1.18 (1.03–1.36) | 0.02 |
| **Sex** | | | | | |
| Female | 9,296/14,395 (64.6) | Reference | — | Reference | — |
| Male | 2,982/4,649 (64.1) | 0.98 (0.92–1.05) | 0.59 | 1.24 (1.15–1.35) | <0.001 |
| **Ethnicity** | | | | | |
| White | 8,147/11,485 (70.9) | Reference | — | Reference | — |
| South Asian | 2,843/4,863 (58.5) | 0.58 (0.54–0.62) | <0.001 | 0.67 (0.62–0.72) | <0.001 |
| Black | 499/1,357 (36.8) | 0.24 (0.21–0.27) | <0.001 | 0.30 (0.26–0.34) | <0.001 |
| Other | 609/1,038 (58.7) | 0.58 (0.51–0.66) | <0.001 | 0.70 (0.61–0.81) | <0.001 |
| Not stated | 180/301 (59.8) | 0.61 (0.48–0.77) | <0.001 | 0.64 (0.50–0.82) | <0.001 |
| **IMD quintile** | | | | | |
| 5 (least deprived) | 3,274/4,597 (71.2) | Reference | — | Reference | — |
| 4 | 2,745/4,010 (68.5) | 0.87 (0.80–0.96) | 0.005 | 0.91 (0.82–1.00) | 0.046 |
| 3 | 2,127/3,302 (64.4) | 0.73 (0.67–0.81) | <0.001 | 0.84 (0.75–0.93) | 0.001 |
| 2 | 2,403/4,085 (58.8) | 0.58 (0.53–0.63) | <0.001 | 0.80 (0.72–0.87) | <0.001 |
| 1 (most deprived) | 1,688/2,940 (57.4) | 0.55 (0.50–0.60) | <0.001 | 0.77 (0.69–0.86) | <0.001 |
| **Occupation** | | | | | |
| Doctor | 1,721/3,001 (57.4) | Reference | — | Reference | — |
| Nurse/midwife/HCA | 4,886/7,815 (62.5) | 1.24 (1.14–1.35) | <0.001 | 1.03 (0.93–1.15) | 0.52 |
| Allied health professional | 953/1,380 (69.1) | 1.66 (1.45–1.90) | <0.001 | 1.40 (1.20–1.62) | <0.001 |
| Administrative/executive | 2,537/3,465 (73.2) | 2.03 (1.83–2.26) | <0.001 | 1.48 (1.32–1.66) | <0.001 |
| Healthcare scientist | 634/871 (72.8) | 1.99 (1.69–2.35) | <0.001 | 1.69 (1.41–2.02) | <0.001 |
| Estates/facilities | 1,399/2,306 (60.7) | 1.15 (1.03–1.28) | 0.015 | 0.93 (0.82–1.05) | 0.27 |
| Other | 148/206 (71.8) | 1.90 (1.39–2.59) | <0.001 | 1.45 (1.04–2.01) | 0.03 |
| **Previous SARS-CoV-2 serology** | | | | | |
| Negative | 7,582/10,314 (73.5) | Reference | — | Reference | — |
| Never tested | 3,800/7,456 (51.0) | 0.37 (0.35–0.40) | <0.001 | 0.46 (0.43–0.50) | <0.001 |
| Positive | 896/1,274 (70.3) | 0.85 (0.75–0.97) | 0.02 | 1.01 (0.88–1.15) | 0.94 |
| **Previous SARS-CoV-2 PCR** | | | | | |
| Negative | 2,311/3,072 (75.2) | Reference | — | Reference | — |
| Never tested | 9,426/15,136 (62.3) | 0.54 (0.50–0.59) | <0.001 | 0.70 (0.64–0.77) | <0.001 |
| Positive | 541/836 (64.7) | 0.60 (0.51–0.71) | <0.001 | 0.71 (0.60–0.85) | <0.001 |
| **Previous COVID-19-related work absence** | | | | | |
| No absence | 7,846/12,489 (62.8) | Reference | — | Reference | — |
| Symptomatic | 2,477/3,698 (67.0) | 1.20 (1.11–1.30) | <0.001 | 1.06 (0.97–1.16) | 0.17 |
| Household or test and trace contact | 1,931/2,727 (70.8) | 1.44 (1.31–1.57) | <0.001 | 1.29 (1.17–1.42) | <0.001 |
| Pregnant | 24/130 (18.5) | 0.13 (0.09–0.20) | <0.001 | 0.21 (0.14–0.34) | <0.001 |

aOR, adjusted odds ratio (adjusted for all variables in the table); COVID-19, coronavirus disease 2019; HCA, healthcare assistant; IMD, Index of Multiple Deprivation; OR, odds ratio; PCR, polymerase chain reaction; SARS-CoV-2, severe acute respiratory syndrome coronavirus 2.

After adjustment, factors associated with uptake of vaccination included older age (age group ≤30 years: aOR 0.48, 95% CI 0.44–0.53 [$p < 0.001$]; age group 51–60 years: aOR 1.19, 95% CI 1.07–1.31 [$p = 0.001$]; both compared to age 41–50 years) and male sex (aOR 1.24, 95% CI 1.15–1.35 [$p < 0.001$]).

HCWs from ethnic minority backgrounds were significantly less likely than their White colleagues to be vaccinated, an effect most marked in those of Black ethnicity (Black: aOR 0.30, 95% CI 0.26–0.34 [$p < 0.001$]; South Asian: aOR 0.67, 95% CI 0.62–0.72 [$p < 0.001$]). We found that vaccination uptake decreased with increasing deprivation (decrease in IMD quintile; test for trend $p < 0.001$).

In comparison to doctors, allied health professionals (aOR 1.40, 95% CI 1.20–1.62 [$p < 0.001$]), administrative and executive staff (aOR 1.48, 95% CI 1.32–1.66 [$p < 0.001$]), and healthcare scientists (aOR 1.69, 95% CI 1.41–2.02 [$p < 0.001$]) were all around 1.5 times more likely to be vaccinated. However, in a sensitivity analysis excluding those with locum/bank contracts (S6 Table), doctors were not less likely than other groups to be vaccinated and were, in fact, more likely than nurses/midwives/HCAs and estates/facilities staff to take up vaccination. Other significant findings remained unchanged.

Staff who had never undergone serology or PCR testing for SARS-CoV-2 were significantly less likely to have been vaccinated than those who had tested negative (serology: aOR 0.46, 95% CI 0.43–0.50 [$p < 0.001$]; PCR: aOR 0.70, 95% CI 0.64–0.77 [$p < 0.001$]). Staff with a history of a previous positive SARS-CoV-2 PCR result were significantly less likely to be vaccinated than those with only negative results (aOR 0.71, 95% CI 0.60–0.85 [$p < 0.001$]). To ensure that this effect was not simply due to these individuals not accessing vaccination due to recent COVID-19 infection, we conducted a sensitivity analysis excluding from the multivariable model individuals who tested positive for SARS-CoV-2 by PCR within the 28 days prior to the vaccine programme starting ($n = 289$). The significant findings remained unchanged.

## Discussion

In this observational analysis in one of the largest and most ethnically diverse populations of HCWs in the UK, we found SARS-CoV-2 vaccine uptake to be significantly lower in those who were younger, female, or living in more deprived areas. We also found that ethnic minority HCWs were significantly less likely to take up vaccination than those of White ethnicity and that this difference was particularly marked for Black HCWs and certain South Asian HCW groups.

We provide real-life observational data from an entire hospital workforce demonstrating that SARS-CoV-2 vaccine uptake is lower in ethnic minority HCWs than those of White ethnicity. This finding aligns with the limited number of published studies conducted on vaccine uptake in HCWs to date [20,21] as well as recent survey studies on SARS-CoV-2 vaccine hesitancy in HCWs in the UK and US [17,31]. It also adds significant weight to emerging data in the general population which also suggest reduced uptake in ethnic minority groups. Our findings closely align with this population-level data as we also demonstrate that those of Black ethnicity were least likely to take up vaccination and that, amongst South Asian ethnic groups, those of Pakistani and Bangladeshi ethnicity were less likely to take up SARS-CoV-2 vaccination than those of Indian ethnicity [32].

Evidence on the specific barriers to COVID-19 vaccination in ethnic minority groups is limited [14]. However, when vaccine uptake is considered more broadly, factors such as a lack of trust in the government or in healthcare systems (e.g., due to unethical and non-ethnically heterogenous research practices in vaccine studies or structural and institutional racism), a lower perception of the risk of COVID-19 or a higher perception of the risk of side effects

from vaccination, and other sociodemographic factors interrelated with ethnicity (educational level, socioeconomic status, and religion) have all been suggested as barriers to vaccine uptake [14,33–35]. Additionally, a recent nationwide survey study of UK HCWs determined that trust in employers and belief in COVID-19 conspiracy theories are important factors in predicting SARS-CoV-2 vaccine hesitancy [17]. Ethnic disparities in vaccine uptake are not unique to COVID-19; this phenomenon was also observed during the 2009 H1N1 influenza pandemic [36,37]. Regardless of the underlying mechanisms, these findings give significant cause for concern, as ethnic minority groups (especially those working in healthcare) are at higher risk of infection with SARS-CoV-2 and adverse outcomes from COVID-19, yet are not taking up this critical preventative intervention [3,5,9,11]. Previous work, including by the UK Scientific Advisory Group for Emergencies (SAGE), suggests that interventions aimed at overcoming barriers to vaccination in ethnic minority groups might involve multilingual, non-stigmatising vaccine endorsements from trusted sources (including information aimed at overcoming religious concerns about the vaccine) and community engagement (utilising trusted sources, e.g., general practitioners, within local communities to respond to concerns about vaccine safety and effectiveness) [14,17,38]. Implementing such interventions urgently is arguably of higher priority in HCWs than in the general population as HCWs represent an important source of health information for ethnic minority communities [14].

Alongside ethnicity, we also found deprivation to be associated with SARS-CoV-2 vaccination uptake, with those living in the most deprived areas being most likely to be unvaccinated. Deprivation has previously been shown to be associated with lower vaccine uptake in the general UK population [39], and, more recently, increased deprivation has been shown to be associated with lower SARS-CoV-2 vaccine uptake in UK HCWs [21]; this may be mediated through many of the same mechanisms discussed in relation to ethnicity above.

In accordance with previous studies [17,31], younger healthcare workers were less likely to be vaccinated than their older colleagues. A likely explanation for this finding is a reduced perception of personal risk of adverse outcomes from COVID-19. However, alongside the obvious greater risk of transmitting infection to more vulnerable individuals, long-term sequelae of COVID-19 (termed 'long-COVID'), which may cause significant morbidity, have been demonstrated to be prevalent even in a young 'low-risk' population [40], suggesting that this cohort may still derive significant personal benefit from SARS-CoV-2 vaccination. A further explanation for this finding is that the vaccination programme was initially targeted at those with risk factors for severe COVID-19 (including those advanced in age), and thus older staff may have had more time and opportunity to be vaccinated compared to their younger colleagues.

We found that doctors were significantly less likely to take up SARS-CoV-2 vaccination than other staff groups (including allied health professionals). However, these findings should be interpreted with caution, as exclusion of individuals with locum/bank contracts resulted in higher uptake amongst doctors. It is possible, therefore, that locum doctors are not taking up vaccination through UHL. This may be due to limited access to trust communications or due to taking up offers of vaccination elsewhere. It should also be noted that this finding is in contrast to the results of 2 previous studies of SARS-CoV-2 vaccine uptake in UK HCWs, which both showed medical staff to be amongst the most likely to take up vaccination [20,21]. Estates and facilities staff also had lower levels of vaccine uptake than many other groups; support staff have been found to have low levels of vaccine uptake previously [20,21,35], and possible explanations for this observation in our cohort include limited access to the email communications regarding vaccination, as well as factors interrelated with occupational role such as educational level, deprivation, and ethnicity.

Though the numbers of pregnant HCWs in our analysis were small, we found that pregnancy was associated with lower odds of being vaccinated. Pregnant HCWs have previously

been shown to be SARS-CoV-2 vaccine hesitant [17]. This finding may be a result of the advice given to pregnant women at the start of the vaccine rollout to delay vaccination until after delivery, on the basis that there was a lack of safety data in this cohort. However, there is now accumulating evidence that SARS-CoV-2 vaccination is safe during pregnancy and also that pregnancy increases the risk of severe COVID-19 [41,42]. Therefore, these findings have important implications for targeting catch-up vaccination programmes.

We also investigated the relationship of previous COVID-19 infection to SARS-CoV-2 vaccine uptake in the UHL population of HCWs. In accordance with a recent study [21], we found that those who were never tested for evidence of current/previous SARS-CoV-2 infection by swab or serology were more likely to be unvaccinated. This is unsurprising given that many of the barriers to vaccination (e.g., mistrust of the healthcare system, 'needle phobia', and low perception of personal risk from COVID-19) may also influence decisions about testing for evidence of current/previous SARS-CoV-2. Those with a history of SARS-CoV-2 PCR positivity were also less likely to take up vaccination. Some of this effect could be mediated by those who were isolating due to a positive swab having no access to vaccination, as well as advice from UHL that those with a positive swab in the last 28 days should avoid vaccination. However, exclusion of individuals testing positive within 28 days prior to the start of the vaccination programme did not change the result, implying the influence of other factors. It is possible that some of those who have had confirmed COVID-19 would be less likely to take up vaccination, believing themselves to have acquired sufficient immunological protection against SARS-CoV-2. This is likely to be true in the short term; however, risk of infection may increase with time since infection, given evidence concerning waning humoral immunity to SARS-CoV-2 and the short-lived immunity after infection with other coronaviruses [43,44]. Therefore, this group may represent an important group to target in subsequent SARS-CoV-2 vaccination drives.

This study has limitations. Although the population is large, data are from a single centre, affecting their generalisability. We only have vaccination data on those who were vaccinated through UHL. HCWs who obtained vaccination through primary care will be coded as unvaccinated in our analysis, although we expect these numbers to be small given that few other vaccination centres were in operation prior to the establishment of the vaccination hubs at UHL. We cannot predict whether HCWs who are currently unvaccinated will take up vaccination in the future; however, the numbers of staff taking up the vaccine over time are falling, implying that most who will accept vaccination have already done so. Furthermore, the predictors of vaccine uptake identified in our surveillance are similar to those in a recent UK-wide survey study of SARS-CoV-2 vaccine hesitancy in HCWs [17] and 2 UK-based studies on SARS-CoV-2 vaccine uptake [20,21], providing reassurance about the generalisability of our results. SARS-CoV-2 PCR testing has been available at other non-UHL centres, and PCR results from HCWs accessing testing via these centres were not available within UHL records; however, given the convenience and availability of PCR testing within UHL, it is likely that the vast majority of staff would have accessed testing via this route. There are other factors that may influence vaccine uptake (e.g., past medical history and educational level) on which we do not have data as we felt this was beyond the scope of an audit, and therefore we cannot adjust for these in our analysis. We are unable to determine which vaccine (Pfizer–BioNTech or Oxford–AstraZeneca) was offered to HCWs, only whether or not they were vaccinated. Therefore, we are unable to determine whether vaccine uptake is associated with the particular vaccine that is offered. This might form a focus of future studies. Despite these limitations, our work has many novel findings that will be of direct relevance to policymakers involved in designing SARS-CoV-2 vaccination programmes, particularly in light of the potential booster vaccination programme for UK HCWs [45].

In summary, we have found that in a population of UK HCWs, those from ethnic minority groups and from more deprived areas, as well as those who are younger, female, or from particular occupational groups, are less likely to take up SARS-CoV-2 vaccination. These findings have major implications for the effective ongoing delivery of SARS-CoV-2 vaccination programmes, including planned booster vaccinations, both in HCWs and in the wider population. Urgent actions should be taken to boost vaccine uptake in the identified groups and to prevent the disparities caused by the COVID-19 pandemic from being allowed to widen further.

## Supporting information

**S1 Checklist. STROBE statement.**
(DOCX)

**S1 Fig. Number and percentage of staff vaccinated over time by age group.** The figure shows the number of staff vaccinated (grey bars) and the cumulative percentage of the total number of staff of each age group vaccinated (coloured lines) each week since the start of the UHL vaccination programme. It should be noted that the first and last time points do not represent complete weeks. There was 1 missing value for date of vaccination, and this observation is excluded.
(TIF)

**S2 Fig. Number and percentage of staff vaccinated over time by sex.** The figure shows the number of staff vaccinated (grey bars) and the cumulative percentage of the total number of staff of each sex vaccinated (coloured lines) each week since the start of the UHL vaccination programme. It should be noted that the first and last time points do not represent complete weeks. There was 1 missing value for date of vaccination, and this observation is excluded.
(TIF)

**S3 Fig. Number and percentage of staff vaccinated over time by Index of Multiple Deprivation.** The figure shows the number of staff vaccinated (grey bars) and the cumulative percentage of the total number of staff of each Index of Multiple Deprivation quintile vaccinated (coloured lines) each week since the start of the UHL vaccination programme. It should be noted that the first and last time points do not represent complete weeks. There were 110 missing values for IMD and 1 missing value for date of vaccination, and these observations are excluded.
(TIF)

**S1 Table. Categorisation of occupational role.** The table shows the categorisation of occupational role from the raw descriptions of occupational role in the Electronic Staff Record.
(DOCX)

**S2 Table. Categorisation of ethnicity.** The table shows the categorisation of ethnicity into the categorical variable used in the main analysis from the descriptions of ethnicity in the Electronic Staff Record.
(DOCX)

**S3 Table. Vaccination status by ethnicity subcategory.** The table shows the proportion of vaccinated and unvaccinated staff of each ethnic group. Classification of ethnic groups was based on the 18 Office for National Statistics categories. Small numbers in some of these groups necessitated collapsing certain categories.
(DOCX)

**S4 Table. Description of cohort excluding locum and bank staff.** The table shows a description of the cohort excluding those with locum or bank contracts by vaccination status. All data are presented as *n* (%). COVID-19, coronavirus disease 2019; HCA, healthcare assistant; IMD, Index of Multiple Deprivation; PCR, polymerase chain reaction; SARS-CoV-2, severe acute respiratory syndrome coronavirus 2.
(DOCX)

**S5 Table. Vaccine uptake in medical staff.** The table shows a description of medical staff by vaccination status. The right-hand columns exclude those with locum or bank contracts. *Values redacted due to the potential for identification. Definitions of roles: foundation year 1 doctor (FY1), the first year of a 2-year training programme for doctors who have just left medical school; foundation year 2 doctor (FY2), the second year of the aforementioned programme; senior house officer (SHO), a doctor in training who has completed the foundation programme and has entered a 'core' training programme such as core medical or core surgical training but has not yet entered speciality training; specialist registrar (SpR), a doctor in training who has entered a speciality training programme; consultant, a hospital doctor who has completed training; trust grade, a doctor who is not in a training programme but is employed by the trust for provision of clinical services (may be of varying grades); medical support staff, roles such as physicians associates and advanced practitioners; general practitioner (GP).
(DOCX)

**S6 Table. Univariable and multivariable analysis of factors associated with SARS-CoV-2 vaccine uptake excluding those with locum or bank contracts.** The table shows unadjusted and adjusted ORs for the association of covariates with the outcome of SARS-CoV-2 vaccine uptake. aOR, adjusted odds ratio (adjusted for all variables in the table); COVID-19, coronavirus disease 2019; HCA, healthcare assistant; IMD, Index of Multiple Deprivation; OR, odds ratio; PCR, polymerase chain reaction; SARS-CoV-2, severe acute respiratory syndrome coronavirus 2.
(DOCX)

## Acknowledgments

Tony Doherty and Michael Dobson assisted with data collection. Matt Archer was involved in establishing the UHL vaccination programme.

## Author Contributions

**Conceptualization:** Prashanth Patel, Charles Goss, David R. Jenkins, Linda Barton, Arthur Price, Nigel J. Brunskill, Kamlesh Khunti, Manish Pareek.

**Data curation:** Christopher A. Martin, Colette Marshall, Prashanth Patel, Charles Goss, David R. Jenkins, Linda Barton, Manish Pareek.

**Formal analysis:** Christopher A. Martin, Manish Pareek.

**Investigation:** Christopher A. Martin, Manish Pareek.

**Methodology:** Christopher A. Martin, Colette Marshall, Manish Pareek.

**Project administration:** Prashanth Patel, Charles Goss, Nigel J. Brunskill, Kamlesh Khunti, Manish Pareek.

**Resources:** Colette Marshall, Prashanth Patel, Charles Goss, Manish Pareek.

**Supervision:** Prashanth Patel, Kamlesh Khunti, Manish Pareek.

**Writing – original draft:** Christopher A. Martin, Manish Pareek.

**Writing – review & editing:** Christopher A. Martin, Colette Marshall, Prashanth Patel, Charles Goss, David R. Jenkins, Claire Ellwood, Linda Barton, Arthur Price, Nigel J. Brunskill, Kamlesh Khunti, Manish Pareek.

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
