## [Editor Report · Decision Letter 0]

17 Feb 2021

Dear Dr Pareek, 

Thank you for submitting your manuscript entitled "Association of demographic and occupational factors with SARS-CoV-2 vaccine uptake in a multi-ethnic UK healthcare workforce: a rapid real-world analysis" for consideration by PLOS Medicine.

Your manuscript has now been evaluated by the PLOS Medicine editorial staff as well as by an academic editor with relevant expertise and I am writing to let you know that we would like to send your submission out for external peer review.

Kind regards,

Dr Raffaella Bosurgi

Executive Editor

PLOS Medicine

---

## [Decision Letter · Decision Letter 1]

4 Aug 2021

Dear Dr. Pareek,

Thank you very much for submitting your manuscript "Association of demographic and occupational factors with SARS-CoV-2 vaccine uptake in a multi-ethnic UK healthcare workforce: a rapid real-world analysis" (PMEDICINE-D-21-00819R1) for consideration at PLOS Medicine. 

[LINK]

In light of these reviews, I am afraid that we will not be able to accept the manuscript for publication in the journal in its current form, but we would like to consider a revised version that addresses the reviewers' and editors' comments. Obviously we cannot make any decision about publication until we have seen the revised manuscript and your response, and we plan to seek re-review by one or more of the reviewers. 

We expect to receive your revised manuscript by Aug 25 2021 11:59PM. Please email us (plosmedicine@plos.org) if you have any questions or concerns.

We look forward to receiving your revised manuscript. 

Sincerely,

Beryne Odeny, 

PLOS Medicine

plosmedicine.org

1) Please revise your title according to PLOS Medicine's style. Your title must be nondeclarative and not a question. It should begin with main concept if possible. Please place the study design (e.g., "A cross-sectional study") in the subtitle (i.e., after a colon). For example, “Factors associated with SARS-CoV-2 vaccine uptake in a multi-ethnic UK healthcare workforce: A cross-sectional study”

2) The Data Availability Statement (DAS) requires revision. For each data source used in your study: 

3) Abstract:

a) Please structure your abstract using the PLOS Medicine headings (Background, Methods and Findings, Conclusions).

b) Please combine the Methods and Findings sections into one section, “Methods and findings”. 

c) Please ensure that all numbers presented in the abstract are present and identical to numbers presented in the main manuscript text.

d) Please include the actual amounts or percentages of relevant outcomes, not just odds ratios.

e) Please include the important dependent variables that are adjusted for in the analyses.

f) Please include p values in addition to 95% CIs.

g) In the last sentence of the Abstract Methods and Findings section, please describe the main limitation(s) of the study's methodology.

4) Author summary - At this stage, we ask that you reformat your non-technical Author Summary. The Author Summary should immediately follow the Abstract in your revised manuscript. This text is subject to editorial change and should be distinct from the scientific abstract. The summary should be accessible to a wide audience that includes both scientists and non-scientists. Please see our author guidelines for more information: https://journals.plos.org/plosmedicine/s/revising-your-manuscript#loc-author-summary.

5) Please conclude the Introduction with a clear description of the study question or hypothesis.

6) The last line of the Introduction section should read, “…and previous COVID-19 infection…”

7) Did your study have a prospective protocol or analysis plan? Please state this (either way) early in the Methods section. 

i) If a prospective analysis plan (from your funding proposal, IRB or other ethics committee submission, study protocol, or other planning document written before analyzing the data) was used in designing the study, please include the relevant prospectively written document with your revised manuscript as a Supporting Information file to be published alongside your study, and cite it in the Methods section. A legend for this file should be included at the end of your manuscript. 

ii) If no such document exists, please make sure that the Methods section transparently describes when analyses were planned, and when/why any data-driven changes to analyses took place. 

iii) In either case, changes in the analysis-- including those made in response to peer review comments-- should be identified as such in the Methods section of the paper, with rationale.

8) Please ensure that the study is reported according to the STROBE guideline, and include the completed STROBE checklist as Supporting Information. Please add the following statement, or similar, to the Methods: "This study is reported as per the Strengthening the Reporting of Observational Studies in Epidemiology (STROBE) guideline (S1 Checklist)." The STROBE guideline can be found here: http://www.equator-network.org/reporting-guidelines/strobe/

9) In the statistical analysis section, please clearly state all the adjustment variables used.

10) In statistical methods, please refer to any post-hoc corrections to correct for multiple comparisons during your statistical analyses. If these were not performed, please justify the reasons. 

11) In the methods section, where appropriate, please consistently provide both 95% CIs and p values in the text and tables.

12) Please specify the statistical test used to derive the p values.

13) Please provide abbreviations in Tables e.g. OR, aOR

14) Discussion section, paragraph 2, line 2, please replace “reduced” with “lower.”

15) Please use the "Vancouver" style for reference formatting and see our website for other reference guidelines https://journals.plos.org/plosmedicine/s/submission-guidelines#loc-references.

a) Please use the PLOS Medicine style reference call outs throughout the text, noting the absence of spaces within the square brackets, e.g., "... approved [2,3]."

b) Please ensure that journal name abbreviations consistently match those found in the National Center for Biotechnology Information (NCBI) databases. https://journals.plos.org/plosmedicine/s/submission-guidelines#loc-references. 

c) Please ensure that six names appear before et al.

16) Please include line numbers in the next draft

Comments from the reviewers:

Reviewer #1: "Association of demographic and occupational factors with SARS-CoV-2 vaccine uptake in a multi-ethnic UK healthcare workforce: a rapid real-world analysis" studies vaccine uptake of almost 20,000 healthcare workers (HCWs) within a large UK hospital workforce (University Hospitals of Leicester NHS Trust). Ethnic minority status, younger age, female sex, increasing deprivation and previous confirmed COVID-19 were all found to negatively correlate with vaccine uptake, from adjusted logistic regression modelling. The analysis is relatively straightforward, and the subject matter of urgent interest.

1. The data extends only up to the 3rd of February (Figure 1), whereas some of the vaccination hubs were established only on the 15th of January. As such, the study appears to cover a relatively short vaccination period. It might be considered to extend the analysis period if possible, or to discuss mitigating factors (e.g. negligible vaccine uptake after Feb 3)

2. In Page 5, it is stated that vaccination was performed initially with the Pfizer-BioNTech vaccine, and subsequently with the Oxford-AstraZeneca vaccine. It might be clarified as to how the vaccines were assigned/whether HCWs were given a choice of which vaccine to take, and whether this might have influenced uptake.

3. In Page 5, it is stated that "Following a short period during which vaccination was limited to those determined to be at highest risk of severe COVID-19, vaccinations were made available to all staff on an ongoing basis". The exact date when vaccination was made available to all, would appear to be very useful in considering the following analysis, and might be stated.

4. In Page 5, it is stated that vaccination appointments had to be booked. It might be stated whether there were ever any scheduling issues (i.e. under-capacity) thoughout the studied period, such that the vaccination of a willing HCW might have had to be postponed.

5. In Page 10, it is stated that "To our knowledge, we are the first to provide real-life data demonstrating that SARS-CoV-2 vaccine uptake is reduced in ethnic minority HCWs"; however, vaccine hesitancy amongst ethnic minorities appears to be relatively well-known (e.g. from "Coronavirus and vaccination rates in people aged 70 years and over by socio-demographic characteristic, England: 8 December 2020 to 11 March 2021", from the Office for National Statistics). Such claims might thus be moderated.

6. It might be considered to provide figures for vaccination over time (as in Figure 1), for other major demographic/risk factors (e.g. age, sex, IMD status, etc)

7. While some speculation about reduced vaccine uptake had been attempted (e.g. "limited access to trust communications" for locum doctors), it would be of interest to briefly survey reasons for declining vaccination if possible, including on possible intent for future vaccination.

Reviewer #2: This is an important paper that provides useful insights for implementing vaccination programs in the UK. I have only few comments for this manuscript:

1) In the introduction, I would like to see more background about the situation of vaccination among HW in other countries. 

2) In the discussion, Indeed, HW should be provided the vaccine freely so it is hard to conclude that there are disparities here. Did this issue occur in the previous epidemic?

3) Please provide more implications of this study. Please also provide more details about the vaccination plan of the UK to address the disparities. 

Reviewer #3: Researchers presented the results of cross-sectional surveillance that described SARS-CoV-2 vaccination uptake among more than 19,000 healthcare workers from the University Hospitals of Leicester NHS Trust. They also determined demographic and socio-economic factors associated with vaccination uptake. The results indicate that vaccine uptake is reduced in ethnic minority healthcare workers, and that is also lower in healthcare workers of younger age, females, and those living in more deprived areas. This is a novel study related to SARS-CoV-2 vaccination uptake; and the manuscript is well written and easy to follow. My comments and suggestions are as follows:

-In the methods section (data collection section in particular), please use subtitles to distinguish independent variables (demographic and socio-economic factors and previous COVID-19 test results from the dependent variable (vaccination uptake). 

-How was ethnicity defined (please provide the ethnicity definition and reference)? What classification the 15 categories of ethnicity are based on?

-In the data analysis section I suggest clarifying that study characteristics are presented based on the vaccination status to help the readers understand what Chi-square test was applied to. Please also provide information on where data were missing. Were data missing only for IMD variable as presented in Table 1, or data for other variables were missing, too?

-Results - I'd suggest leaving only percentages and leaving out counts, as both counts and percentages are already available in Table 1 (e.g. saying "Of the 19,044 HCW in the cohort, 64.5% had received ....."This will also make the presentation of the results consistent. 

-In tables 1, S3, S4, S5 I suggest deleting '%' in each row and stating that all data are presented as n (%). Considering Chi-square was used, the researchers may wish to add a Significance column to table 1 and present p values that reflect results of the Chi-square tests. 

-Table 2 - I suggest defining aOR i.e. what adjustment it pertains to. In the same table, in column 2, all '%' could be deleted, as it states at the top of the table that all numbers present N vaccinated/N total (%). In the title of table 2, 'ke' is missing in the word 'uptake'

-Discussion: please compare your results with a recently published study of Woolf et al https://www.thelancet.com/journals/lanepe/article/PIIS2666-7762(21)00157-5/fulltext Please also check for other studies published on a similar topic, such as the recent one of Kuter et al https://pubmed.ncbi.nlm.nih.gov/33632563/

[LINK]

---

## [Decision Letter · Decision Letter 2]

13 Sep 2021

Dear Dr. Pareek,

Thank you very much for re-submitting your manuscript "Factors associated with SARS-CoV-2 vaccine uptake in a multi-ethnic UK healthcare workforce: A cross-sectional study" (PMEDICINE-D-21-00819R2) for review by PLOS Medicine.

I have discussed the paper with my colleagues and the academic editor and it was also seen again by xxx reviewers. I am pleased to say that provided the remaining editorial and production issues are dealt with we are planning to accept the paper for publication in the journal.

[LINK]

We look forward to receiving the revised manuscript by Sep 20 2021 11:59PM.   

Sincerely,

Beryne Odeny, 

Associate Editor 

PLOS Medicine

plosmedicine.org

Requests from Editors:

1) Please consider revising your title to “SARS-CoV-2 vaccine uptake in a multi-ethnic UK healthcare workforce: A cross-sectional study.” Apologies for requesting a second revision

2) Please add the following statement to the Methods: "This study is reported as per the Strengthening the Reporting of Observational Studies in Epidemiology (STROBE) guideline (S1 Checklist)."

3) Thank you for providing your STROBE checklist. Please replace the page numbers with paragraph numbers per section (e.g. "Methods, paragraph 1"), since the page numbers of the final published paper may be different from the page numbers in the current manuscript.

4) Thank you for updating your references. Please ensure there is a space between in-text reference call outs and the preceding word, and the full stop is placed after the call out. For example, “…approved [2,3].”

Comments from Reviewers:

Reviewer #1: We thank the authors for addressing our previous comments, and have no further queries.

Reviewer #3: I thank the researchers for addressing all comments and suggestions raised by the editorial team and reviewers. No further comments.

[LINK]

---

## [Editor Report · Decision Letter 3]

23 Sep 2021

Dear Dr Pareek, 

On behalf of my colleagues and the Academic Editor, Dr. Aaron S. Kesselheim, I am pleased to inform you that we have agreed to publish your manuscript "SARS-CoV-2 vaccine uptake in a multi-ethnic UK healthcare workforce: A cross-sectional study" (PMEDICINE-D-21-00819R3) in PLOS Medicine.

PRESS

Sincerely, 

Beryne Odeny 

Associate Editor 

PLOS Medicine